# OpenReview forum: "Gradient Cuff: Detecting Jailbreak Attacks on Large Language Models by Exploring Refusal Loss Landscapes"
_NeurIPS.cc/2024/Conference — NeurIPS 2024 poster_

### Official Review · Reviewer_yW4P · 2024-06-21

**Soundness:** 2
**Presentation:** 3
**Contribution:** 2
**Rating:** 5
**Confidence:** 4

**Summary:**

This paper analyzes the loss landscape of the refusal loss in the jailbreaking problem and proposes a method to detect jailbreaking input. This method has two steps: 1) sample multiple outputs and vote on the results, and 2) take the gradient with respect to the input and measure the gradient norm. Overall, the proposed methods are effective.

**Strengths:**

- The proposed methodology can effectively detect jailbreaking input.
- The paper is well-organized and the overall writing is good.
- The evaluation covers a number of attacks and adaptive attacks.

**Weaknesses:**

- **The main weakness** is that the proposed method requires $N · (P + 1)$ queries. In the paper, the authors select $N=P=10$ , which means it requires more than 100 queries for one sentence. Even though the authors claim that one can use batch inference, the memory cost makes the proposed method unrealistic in practice. Step 1 of the proposed method is trivial and not sound due to its computing efficiency. The paper does not compare the time and memory efficiency with other baselines.

- The presentation of the experimental result is not good. The paper puts Figure 2 and Table 1 instead of the full results (Table A1 and Table A2) in the main body. I would suggest combining Table A1 and Table A2 and putting them in the main body. Additionally, Figure 2 is a bit unclear and insufficient, e..g., what the blue line indicates?

- Equations 4 and 5 are not very necessary because the message is just that the authors add the same perturbation for all embeddings. Theorem 1 is somehow redundant.

- The considered jailbreak keywords are much less than those used in GCG's paper.

**Questions:**

- In sec 4.5 Utility Analysis, the author compares Vicuna-7B-V1.5 & Vicuna-7B-V1.5 with Gradient Cuff, why the metric is 'Average answer accuracy' instead of 'FDR'? What we actually care about is whether a model will provide an answer to a benign sentence, if both models provide output, then such outputs should be the same. What is the meaning of ``By setting a 5% FPR on the validation dataset''?
- Could the author compare or discuss the proposed method against [2,3,4]?
- Is it possible to apply the proposed defending method in closed-source GPT?
- How to sample in a batch with the same temperature?
- Why the considered jailbreak keywords are much less than those used in GCG's paper [1].

### Ref
* [1] Universal and Transferable Adversarial Attacks on Aligned Language Models
* [2] Safedecoding: Defending against jailbreak attacks via safety-aware decoding
* [3] Single-pass detection of jailbreaking input in large language models
* [4] RAIN: Your language models can align themselves without finetuning

**Limitations:**

Currently, I don't see much clear discussion on the limitations even though the authors claim they are in sections 4.4 and 4.5. The discussed limitations hold for all jailbreaking defenses instead of only the proposed methods.

---

> ### Author Rebuttal · Authors · 2024-08-05
>
> **Weakness 1**: The paper does not compare the time and memory efficiency with other baselines.
>
> **R**: We follow the setting in Appendix A.15 to evaluate different methods’ average running time and the required largest GPU memory on Vicuna-7b-V1.5.
> ||Time(s)|Memory(GB)|TPR|
> |---|---|---|---|
> |w/o defense|0.41|26.10|0.233|
> |PPL | 0.44|26.10|0.401|
> |SmoothLLM|4.24|42.75|0.413|
> |Gradient Cuff|4.59|42.36|0.743|
>
> From the table, we can find that
>
> **(1)** PPL has the shortest running time and lowest memory need but is outperformed by Gradient Cuff by a large margin (0.401vs0.743).
>
> **(2)** SmoothLLM has a similar time and memory need to Gradient Cuff, but a much worse performance (0.413vs0.743)
>
> We will include this time-memory comparison in the revised version of the paper.
>
> ***
>
> **Weakness 2**: The presentation of the experimental result is not good. Additionally, Figure 2: what does the blue line indicate?
>
> **R**: We thank the reviewer for pointing out this. We will improve our presentation following the reviewer's suggestion.  In Figure 2, the blue line is the indicating line to make it clear what method each point represents.
>
> ***
>
> **Weakness 3**: Equations 4 and 5 are not very necessary. Theorem 1 is somehow redundant.
>
> **R**:Thank the reviewer’s suggestion. We want to use Equations 4 and 5 to show that the operation of adding the same perturbation for all embeddings is equivalent to adding the perturbation to the sentence embedding of the input query if the sentence embedding is acquired by applying mean-pooling. These two equations are very important for formalizing our method into a mathematical operation and then derive the theorem 1.
>
> As for theorem 1, this theorem showed that we can decrease the estimation error by improving both P and N, which provides theoretical evidence for results in Appendix A.13 and Appendix A.14 which point out the way to continuously enhance Gradient Cuff.
>
>
>
> ***
>
> **Weakness 4 and Question 5**: Why the considered jailbreak keywords are much less than those used in GCG's paper.
>
> **R**: We slightly change the keyword list in the GCG paper for two reasons:
>
> (1)we found the keyword list is somehow redundant as many different keywords will appear at the same time. For example, the model often responds to malicious queries using “I apologize but as an”, so we just need to pick one between “as an” and “I apologize” when designing the keyword list.
>
> (2)Some keywords will cause false accusations. for example, GCG’s original keyword list contains “hello” so all the model outputs containing “hello” will be regarded as model rejection, which is not reasonable.
>
> We do realize that using keyword matching as the computation method of ASR will introduce instability, so we decided to include a different evaluation method in the revised version of the paper.  We choose two other metrics to compute the ASR:
>
> (1) GPT-4: we designed the system prompt of GPT-4 so that the GPT4 would assist in checking whether the model response implies jailbroken.
>
> (2) LLaMAGuard: we input the user query and the model response to llamaguard. The llamaguard model would output “unsafe” if the model response implies jailbroken.
>
> We compared Gradient Cuff with baseline methods under these two evaluation methods and the results are shown below:
>
> |  |  | ASR |  |
> |---|---|---|---|
> |  |  | GPT-4 Metric | LLaMA-Guard Metric |
> | LLaMA | Gradient Cuff | 0.0279 | 0.0229 |
> |  | SmoothLLM | 0.0192 | 0.0188 |
> |  | PPL | 0.0463 | 0.0279 |
> |  | wo/defense | 0.1621 | 0.1296 |
> | Vicuna | Gradient Cuff | 0.1296 | 0.1171 |
> |  | SmoothLLM | 0.2354 | 0.2408 |
> |  | PPL | 0.3200 | 0.2854 |
> |  | w/o defense | 0.4637 | 0.4254 |
>
> We can conclude from the table that the Gradient Cuff still offers great advantages under new evaluation methods. On vicuna, Gradient Cuff outperforms the best baselines by a large margin (0.1296 vs 0.2354 under GPT4 evaluation and 0.1171 vs 0.2408 under llama guard evaluation). On llama, Gradient Cuff achieves comparable performance with the best baselines, 0.0279vs0.0192 under GPT-4 evaluation and 0.0229vs0.0188 under llama guard evaluation.
>
> ***
>
> **Question 1**: In sec 4.5, why the metric is 'Average answer accuracy'? What is the meaning of ``By setting a 5% FPR on the validation dataset''?
>
> **R**: We use the average answer accuracy as the metric, which is the standard evaluation metric of the MMLU benchmark.
>
> Recall that we collect a bunch of benign prompts and split them into test and validation subsets. Our Gradient Cuff needs to specify the detection threshold, and we use the FPR on the validation subset as the criteria when adjusting the threshold. By setting a 5% FPR on the validation dataset, We want to ensure that the utility degradation would be constrained to an acceptance range.
>
> ***
> **Note**: Due to the space limit, we cannot answer all of this reviewer's questions in one rebuttal (we need 5542 characters to answer the following questions but we only have 1247 characters left in this response box.), the rest of the questions are listed below:
>
> **Question 2**: Could the author compare or discuss the proposed method against [2,3,4]?
>
> **Question 3**: Is it possible to apply the proposed defending method in closed-source GPT?
>
> **Question 4**: How to sample in a batch with the same temperature?
>
> we also find these 3 questions touching upon the general discussion of our work and we hope every reviewer can read our clarification on these 3 questions, so we put our response to these 3 questions in the general rebuttal space: https://openreview.net/forum?id=vI1WqFn15v&noteId=LDZ08CLDUs

---

> > ### Comment · Reviewer_yW4P · 2024-08-12
> > **Thank the authors for the explanation**
> >
> > Thank the authors for the explanation. I have read the comments and rebuttal, I will maintain the score due to the main weakness of using $N*(P+1)$ queries.

---

> > > ### Author Response · Authors · 2024-08-13
> > > **Response on the extra inference cost**
> > >
> > > We thank the reviewer for the feedback on our rebuttal.
> > >
> > > We acknowledge that current strong defenses (e.g., SmoothLLM) and our proposed defense come with an increased inference cost, but our results on multiple jailbreak guardrails and LLMs suggest an inevitable tradeoff between safety and inference costs: Because users may have less incentive to use a model/service if it does not have proper safety guardrails. Moreover, existing baselines are not as effective as ours, no matter they are lightweight defenses or they have similar running costs to us.
> > >
> > > Moreover, as jailbreak detectors are a frontier and active research area, we believe Gradient Cuff can inspire future studies, and its inference efficiency can be improved by future works.
> > >
> > > If the reviewer finds the concerns are mostly addressed, we hope the reviewer will adjust the rating accordingly.

---

### Official Review · Reviewer_RDin · 2024-07-08

**Soundness:** 3
**Presentation:** 2
**Contribution:** 2
**Rating:** 5
**Confidence:** 3

**Summary:**

This paper introduces the concept of refusal loss function for detecting language model jailbreaking attacks. The refusal loss is defined as $1$ minus the refusal rate, and it is observed that on malicious instructions, the refusal loss tends to have a smaller value and a larger gradient norm. Based on this observation, a two-stage defense algorithmic framework called "Gradient Cuff" is proposed to detect jailbreaking attempts by setting thresholds on these two quantities. Experimental results on Vicuna-7b-v1.5 and Llama2-7b-chat demonstrate the effectiveness of the proposed method against baseline defenses, including the perplexity filter, the erase-and-check method, SmoothLLM, and Self Reminder.

**Strengths:**

The paper's contribution lies in the investigation of the refusal loss landscape, which is a novel and intriguing aspect. While the definition of the refusal loss itself is not new, the exploration of its properties and behavior in the context of language model jailbreaking attacks is original and valuable.

The proposed Gradient Cuff method has several advantages over existing baseline defense methods. Firstly, it can detect jailbreak attacks in multiple forms, whereas methods like the perplexity-based filter are mostly effective against suffix-based attacks. Secondly, Gradient Cuff allows for direct control over the false positive rate (FPR) versus true positive rate (TPR), which is not easily achievable by most existing defense methods. This feature provides a useful trade-off between detection accuracy and false alarm rates.

The ablation experiments conducted in the paper are comprehensive and effectively illustrate the usefulness of the two proposed stages separately. The comparison against baseline defenses also showcases the advantages of Gradient Cuff over existing methods. Overall, the paper presents a promising approach to detecting language model jailbreaking attacks and contributes to the ongoing efforts to improve the safety and security of language models.

**Weaknesses:**

My concern of weaknesses are as follows.

I. A discrepancy between the 2-stage introduction of Gradient Cuff in the main text and its description in Appendix A.5, where an additional 3rd step involving sampling and rejection via the JB function is considered. This operation caught my attention because (1) it implies an extra layer to the introduction of Gradient Cuff in the final step, and (2) no ablation on the JB function was provided to study its impact on Gradient Cuff's performance. Prior research (e.g., [1], Figure 2) has demonstrated the instability of string matching, which is the JB function used throughout the paper. Therefore, an ablation on this component (if implemented throughout the paper as the 3rd stage of Gradient Cuff for TPR computation) should be taken into consideration.

II. The experimental setup is confusing across different tables and figures. In line 229, it is stated that σ (FPR) is set to 5%. However, the full result of Figure 2 (Table A1) contradicts this claim - one could also tell this from Figure 2 directly. This also makes the comparison in Table 3 invalid, since the numbers appearing without adaptive attack do not have σ (FPR) as 5%. The numerical description in line 259 on improvements is also invalid.

III. Llama2-chat system prompt usage. In section 4.1, it is claimed that "As for the system prompt, we use the default setting provided in the fastchat repository." However, fastchat does not provide the default Llama2-chat system prompt by default (see https://github.com/lm-sys/FastChat/pull/2162), which is **not common** for jailbreak evaluations (see e.g., [1] [2] [3]). This is partially supported by the numbers in Figure A1 (a), where the GCG attack success rate on Llama2-chat is 0.694, which is significantly higher than reported in previous works [1][2].

[1] Mazeika et al, HarmBench: A Standardized Evaluation Framework for Automated Red Teaming and Robust Refusal.

[2] Zou et al, Universal and Transferable Adversarial Attacks on Aligned Language Models.

[3] Chao et al, JailbreakBench: An Open Robustness Benchmark for Jailbreaking Large Language Models.

**Questions:**

I have the corresponding questions following the weaknesses section.

I. How much does the 3rd JB indicator layer functions for Gradient Cuff, and what happens if we ablate this layer with other indicator candidates (e.g., with Llama Guard)? Will Gradient Cuff still offer great advantages, or the most contribution will be attributed to the JB filter?

II. Should the FPR in Figure 2 and Table A1 be exactly 5%?

III. What happens to Table 2 if the correct Llama2-chat system prompt is used? Do we still see significant increments over TPR when we incorporate stage 2 of Gradient Cuff into consideration?

**Limitations:**

Yes

---

> ### Author Rebuttal · Authors · 2024-08-05
>
> **Weakness 1 and Question 1**: A discrepancy between the 2-stage introduction of Gradient Cuff in the main text and its description in Appendix A.5, where an additional 3rd step involving sampling and rejection via the JB function is considered. How much does the 3rd JB indicator layer functions for Gradient Cuff, and what happens if we ablate this layer with other indicator candidates (e.g., with Llama Guard)? Will Gradient Cuff still offer great advantages, or the most contribution will be attributed to the JB filter?
>
> **R**: We thank the reviewer for pointing out this issue that may cause misunderstanding. We believe this is a misunderstanding, due to the discrepancy in implementing and evaluating jailbreak detectors.  In fact, Appendix A.5 just illustrates how we compute the Attack Success Rate(ASR) for Gradient Cuff (That is, the evalaution phase). So the 3rd step isn’t the component of the Gradient Cuff. It is just defining the evaluation metric that we reported, by checking how many jailbreak prompts that passed the Gradient Cuff filtering can successfully attack the protected LLM.
>
> We agree with the reviewer that the computation method for ASR may influence the evaluation and we should also consider other candidates to compute the ASR. To address the reviewer’s concern, We choose two other metrics to compute the ASR:
>
> **GPT-4**: We follow PAIR to design the system prompt of GPT-4 so that the GPT4 would assist in checking whether the model response implies jailbroken.
>
> **LLaMAGuard**: We input the user query and the model response to llama guard. The llama guard model would output “unsafe” if the model response implies jailbroken.
>
> We compared Gradient Cuff with baseline methods under these two evaluation methods and the results are shown below:
> |  |  | GPT-4 ASR | LLaMA-Guard ASR | Utility |
> |---|---|---|---|---|
> | LLaMA | Gradient Cuff | 0.0279 | 0.0229 | 0.3780 |
> |  | SmoothLLM | 0.0192 | 0.0188 | 0.2706 |
> |  | PPL | 0.0463 | 0.0279 | 0.4107 |
> |  | wo/defense | 0.1621 | 0.1296 | 0.4517 |
> | Vicuna | Gradient Cuff | 0.1296 | 0.1171 | 0.4621 |
> |  | SmoothLLM | 0.2354 | 0.2408 | 0.3032 |
> |  | PPL | 0.3200 | 0.2854 | 0.4867 |
> |  | w/o defense | 0.4637 | 0.4254 | 0.4867 |
>
> We can conclude from the table that the Gradient Cuff still offers great advantages under new evaluation methods. On vicuna, Gradient Cuff outperforms the best baselines by a large margin (0.1296 vs 0.2354 under GPT4 evaluation and 0.1171 vs 0.2408 under llama guard evaluation). On llama, Gradient Cuff achieves comparable performance with the best baselines, 0.0279vs0.0192 under GPT-4 evaluation and 0.0229vs0.0188 under llama guard evaluation.
>
> We also include the utility of different baselines in this table, from this table we can see that
> PPL and Gradient Cuff both keep a good utility while SmoothLLM will significantly degrade the utility of the protected LLM.  The dramatic utility degradation of SmoothLLM is due to its random modification to the original user query which has greatly changed the semantics of the original query. This is also the explanation for why SmoothLLM can get an ASR close to Gradient Cuff on LLaMA-2-7B-Chat when evaluated by GPT-4 and LLamaGuard: many malicious queries have lost the maliciousness after SmoothLLM’s modification, so even if the LLM doesn't refuse to answer them, the generated response is harmless.
>
> We will provide an analysis of these different evaluation methods in the revised version of the paper.
>
> ***
>
> **Weakness 2 and Question 2**: Should the FPR in Figure 2 and Table A1 be exactly 5%?
>
> **R**: We believe this is a misunderstanding of the FPR on the validation set (σ that is set to be 5%) and the FPR of the test set. We mentioned in section 4.1 that we collect a bunch of benign user queries and split them into a validation subset and a test subset. We used the validation subsets to determine the detection threshold, and our detector does not have access to the test set. In line 229, we said that we set the σ (FPR) on the validation set to be 5% to control the FPR on the test set. As the test set and the validation set are independently identical, the FPR(test set) and FPR(validation) may be very close but still have small differences. In table A1, we can see on LLaMA, FPR(val) =5% and FPR(test set) = 2.2%. On Vicuna, FPR(val) =5% and FPR(test set) = 3.14%.
>
> ***
>
> **Weakness 3 and Question 3**: Llama2-chat system prompt usage.
>
> **R**: We agree with the reviewer that the system prompt would influence the performance. However, we used fschat(0.02.23) in all our experiments to keep consistent with the GCG repo(see https://github.com/llm-attacks/llm-attacks?tab=readme-ov-file#installation). Upon checking, fschat of this version provides the chat system prompt for LLaMA-2.
>
> The increased  ASR on llama compared with the reported value may come from longer training steps. GCG paper indicates that longer training steps would make the generated suffix more powerful. The reported value is corresponding to 500 training steps. We want to test our method on more powerful jailbreak attacks so we trained the GCG for 1000 steps, which we indicated in the appendix A.2. So we think a higher ASR (w/o defense) is reasonable.

---

> > ### Comment · Reviewer_RDin · 2024-08-09
> > **Response to rebuttal**
> >
> > I would like to extend my gratitude to the authors for their clarifications and for conducting additional experiments. I apologize for my previous misunderstanding regarding the FPR. I appreciate the updated results demonstrating that Gradient Cuff continues to hold its advantages over the (newly included) baselines, even as the jailbreak judge varies.
> >
> > At this moment, I have no significant concerns. After reviewing the rest of the rebuttal and the general response, I agree with reviewer GFwB that while batch inference is beneficial, the computational cost of deploying Gradient Cuff could impact its practicality. Given the effectiveness of the proposal across multiple LLMs and the aforementioned limitation, I have raised my score to 5.

---

> > > ### Author Response · Authors · 2024-08-10
> > > **Thanks for the score increase and response on the pointed-out limitations.**
> > >
> > > We thank the reviewer for the constructive comments and the score increase! The new results suggested by the reviewer certainly strengthen our contributions.
> > >
> > > While we acknowledge that current strong defenses (e.g., SmoothLLM) and our proposed defense come with an increased inference cost, our results on multiple jailbreak guardrails and LLMs suggest an inevitable tradeoff between safety and inference costs (more specifically, our results on the appendix A.14 showed that existing baselines cannot outperform us even though they scale up their performance by increasing their inference cost to a close level with us). Nonetheless, upon recognizing the effectiveness of our proposed defense, we believe our paper will motivate future studies toward the direction of developing more cost-efficient jailbreak detectors and putting more emphasis on AI safety.

---

### Official Review · Reviewer_GFwB · 2024-07-10

**Soundness:** 3
**Presentation:** 3
**Contribution:** 2
**Rating:** 5
**Confidence:** 4

**Summary:**

The authors propose Gradient Cuff that uses the gradient norm for response y being a non-refusal (using their refusal loss) towards the prompt x to detect jailbreak attacks. Since the designed loss is non-differentiable (along with the generation process), the authors propose to use a zeroth order gradient estimate. Here the gradient is estimated via multiple evaluations using slightly perturbed input embeddings. The authors show that Gradient Cuff is increasing the refusal rate significantly for state-of-the-art jailbreak attacks while maintaining reasonable utility.

**Strengths:**

1. The adaptive evaluation makes clear that the increased refusal rate is meaningful
1. The authors highlight that for many prompts resulting from jailbreak attacks, (a) the attacked model sometimes refuses and sometimes answers to the request and (b) the refusal loss gradient towards the adversarially crafted prompt is of large norm if the attack was successful
1. The method Gradient Cuff is simple yet effective
1. The false positive rate of Gradient Cuff is expected to be low in many domains since most LLMs only rarely respond with phrases like "I am not able to" etc.

**Weaknesses:**

1. The defense may be considered costly. In the worst case, the Gradient Cuff requires N (P + 1) model evaluations (10 * 11 = 110 in the experiments). Although this cost is somewhat mitigated by a two-step procedure. The argument of batching is somewhat questionable, considering that the LLM operator could batch multiple user requests otherwise.
1. Only two LLMs are evaluated that are both Llama-based. It would make the paper more convincing if the gradient phenomenon is shown to hold for a large number of LLMs.
1. The utility evaluation is rather shallow, only focusing on multiple languages.

**Questions:**

1. What dataset is used for tuning the false positive rate? How well does this parameter generalize to slightly different settings? Can the authors also report, e.g., AUC scores for independence of a specific threshold?
1. What is the reason for the (costly) zero-th-order gradient estimates? I imagine that one can also use backprop at each generation step that yields certain jailbreak keywords (token), using the log-likelihood of this keyword (token).

**Limitations:**

Limitations are sufficiently addressed

---

> ### Author Rebuttal · Authors · 2024-08-05
>
> **Weakness 1**: The argument of batching is somewhat questionable, considering that the LLM operator could batch multiple user requests otherwise.
>
> **R**: We thank the reviewer for pointing out this issue that may cause misunderstanding. We need to argue that our method is designed for model owners deploying safety guardrails, instead of a user calling an LLM operation. This setting makes our proposed batch inference feasible. One just needs to put multiple user queries into a batch and get a batch of model outputs as LLM’s responses to the queries. A sample code is shown below:
>
> ```python
> def batch_inference(prompt_list, model_name):
>    #prompt list contains multiple user query sentecnes
>    model = AutoModelForCausalLM.from_pretrained(model_name).cuda()
>    tokenizer = AutoTokenizer.from_pretrained(model_name)
>    inputs=tokenizer(prompt_list,return_tensors="pt",padding=True)
>    inputs={k:v.to(model.device) for k,v in inputs.items()}
>    with torch.no_grad():
>        input_ids = inputs["input_ids"]
>        attention_mask = inputs["attention_mask"]
>        prompt_length=input_ids.shape[-1]
>        outputs = model.generate(
>            input_ids = input_ids,
>            attention_mask= attention_mask,
>            do_sample = True
>        )
>        output_text = tokenizer.batch_decode(outputs[:,prompt_length:], skip_special_tokens=True)
>    return output_text
> ```
> We will provide a sample code and more explanations in the future revised version to avoid misunderstanding.
>
> ***
>
> **Weakness 2**: It would make the paper more convincing if the phenomenon is shown to hold for more LLMs.
>
> **R**: We agree with the reviewer that we should test on more LLMs. We selected Qwen2-7B-Instruct and Gemma-7b-it to verify Gradient Cuff’s effectiveness.
>
> As the jailbreak prompts generation (GCG, PAIR, AutoDAN, TAP) is time-consuming and we only have one week to prepare for the rebuttal, we can’t generate jailbreak prompts for the two new models. We choose to test against Base64 attacks, which is a model-agnostic jailbreak attack method. We also tested Gemma and Qwen2 against GCG attacks transferred from Vicuna, which the authors of GCG claimed to have good transferability. The results are summarized in the following table. (the metric is the refusal rate, higher is better)
>
> |  | Attack | w/o defense | gradient cuff | ppl | smoothllm |
> |---|---|---|---|---|---|
> | gemma-7b-it | GCG(Vicuna-7b-v1.5) | 0.89 | 0.91 | 0.97 | 0.91 |
> |  | Base64 | 0.01 | 0.66 | 0.01 | 0.00 |
> |  | Average | 0.45 | **0.79** | 0.49 | 0.45 |
> | qwen2-7b-instruct | GCG(Vicuna-7b-v1.5) | 0.76 | 0.97 | 1.00 | 0.77 |
> |  | Base64 | 0.03 | 1.00 | 0.03 | 0.00 |
> |  | Average | 0.40 | **0.99** | 0.52 | 0.39 |
>
> The results in this table show that our Gradient Cuff can achieve superior performance on non-LLaMA-based models like Gemma and Qwen2, outperforming SmoothLLM and PPL by a large margin. We believe our method can generalize to other aligned LLMs as well.
>
> Though we cannot get the jailbreak prompts for new models due to the time limit of the rebuttal and the limit of computing resources, we've been running these attacks and will provide updates once they are done.
>
> ***
>
> **Weakness 3**: The utility evaluation is rather shallow, only focusing on multiple languages.
>
> **R**: We used MMLU Average Answer Accuracy as the utility metric, which is a common and popular metric used in many mainstream LLM leaderboards.  The MMLU contains 57 tasks across various topics, including elementary mathematics, US history, computer science, and law. MMLU is a valuable tool for evaluating the performance of language models and their ability to understand and generate language in various contexts.
>
> ***
>
> **Question 1**: What dataset is used for tuning the false positive rate?  Can the authors also report, e.g., AUC scores for independence of a specific threshold?
>
> **R**: We thank the reviewer for this question. We mentioned in section 4.1 that we collect a bunch of benign user queries and split them into a validation subset and a test subset. We used the validation subsets to determine the detection threshold thus adjusting the FPR evaluated on the test subset.
>
> Moreover, we need to argue that we didn’t use AUC as a metric because only Gradient Cuff and PPL can provide such adjustable thresholds so we cannot compute AUCs for other baselines like SmoothLLM.
>
> Below we show the AUCs of Gradient Cuff and PPL:
> |  | PPL | Gradient Cuff |
> |---|---|---|
> | LLaMA-2-7B-Chat | 0.7847 | 0.9692 |
> | Vicuna-7B-V1.5 | 0.5227 | 0.9273 |
>
> ***
>
> **Question 2**: What is the reason for the (costly) zero-th-order gradient estimates? I imagine that one can also use backprop at each generation step that yields certain jailbreak keywords (token), using the log-likelihood of this keyword (token).
>
> **R**: We have introduced the motivation of Gradient Cuff in the introduction and the method section and we do agree with the reviewer that motivation needs more explanation. Our refusal function tries to use the empirical probability to estimate the LLM’s theoretical probability of rejecting the query by sampling multiple model responses for it and computing the refusal ratio. The log-likelihood of keyword tokens cannot be used as the proxy of the refusal probability as:
>
> （1）The rejection keywords may not always appear at the beginning of a sentence representing rejection in real-world cases.
>
> （2）The log-likelihood of the keyword is just the loss of the next-word-prediction, it cannot be used to model the appearance probability of the keyword. For example, even some model hallucinations will have a large log-likelihood as long as it is readable and fluent.

---

> > ### Comment · Reviewer_GFwB · 2024-08-08
> > **Response to rebuttal**
> >
> > I thank the authors for the detailed explanations and the effort in running additional experiments!
> >
> > Most concerns are resolved; however, the high computational cost remains.
> >
> > I am not doubting that batch processing is possible or may yield gains. However, whoever runs the model will ~100x their compute cost. Instead of batching the same request for gradient cuff, whoever runs the model, could otherwise batch different prompts/requests instead.

---

> ### Author Response · Authors · 2024-08-08
> **Response on the extra inference cost**
>
> We thank the reviewer for the feedback. We now better understand the reviewer's comment related to inference cost. We agree with the reviewer that this defense comes with some extra inference cost (although with batch inference the time cost would be only 2x~11x if the extra query used by the defense is 110x, see our analysis in appendix A.15).
>
> Indeed, one can save the extra queries used to deploy the defense to serve more users or one user's multiple requests. However, we believe the extra query budget should still be allocated to implement the defense, because users may have less incentive to use a model/service if it does not have proper safety guardrails.
>
> The increased inference cost of Gradient Cuff does bring much-improved jailbreak defense performance and we think this is an inevitable tradeoff because,  as we discussed in the response to the reviewer yW4P proposed weakness1(https://openreview.net/forum?id=vI1WqFn15v&noteId=sWvR1tuNOB), our defense gets the most superior trade-off compared with existing baselines. Smoothllm, the most powerful baseline under our evaluation, has a similar inference cost to us but the performance is far behind us. PPL, a very lightweight defense that almost does not bring extra inference cost, has a poor jailbreak defense capability and only works on specific types of attacks.

---

> ### Author Response · Authors · 2024-08-12
> **thank the reviewer for score increase**
>
> We thank the reviewer for the score increase!
>
> We think the reviewer's constructive suggestions do point out new directions to improve this work, we will update our discussion in the revised version of our paper.

---

### Official Review · Reviewer_AWBZ · 2024-07-11

**Soundness:** 3
**Presentation:** 3
**Contribution:** 3
**Rating:** 6
**Confidence:** 3

**Summary:**

In this paper, the authors proposed Gradient Cuff, a novel jailbreak detection method based on the refusal loss landscape. Specifically, 1) the refusal loss is defined as the probabiliy for the LLM to generate a refusal as response; 2) the authors further made the observation that the refusal loss tend to be more precipitous for malicious queries than benign queries. Accordingly a two-stage detector surfaces: 1) rejecting the queries with the refusal loss less than 0.5; 2) rejecting the queries with the gradient norm of the refusal loss larger than a threshold. In practice, Gradient Cuff perfoms multiple independent probabilistic inferences on a query to estimate refusal loss as the frequency to observe refusal related keywords in the reponses; it uses zero order estimation to approximate the gradient with random purtabations to the input token embeddings; the natural refusal rate is measured with knowingly benign queries to facilitate the decision of the gradient norm threshold. This detection methods was evaluated together with baseline methods like Smooth-LLM and Self-Reminder on a subset of the AdvBench dataset agaisnt various jailbreaking attacks including AutoDAN and PAIR. Results showed that Gradient Cuff yields both a higher refusal rate for malicious queries and lower refusal rate for benign queries, indicating that it provides better defence. Adaptive attack experiments are also done where the attacks are launched when Gradient Cuff is already in place. The results were also in favor of Gradient Cuff.

**Strengths:**

+ The authors did a good job illustrating the refusal loss with mathematicl formulations as well as plots.
+ The paper comes with a proof of the error in the gradient norm estimation as well as a verification through empirical results.

**Weaknesses:**

Gradient Cuff requires actually analysing the harmfulness of the generated response. Although the authors claim that the overhead due to Gradient Cuff can be even lower than some of the baselines, it is still a significant difference, so the comparison with them are not entirely fair.

**Questions:**

+ As Gradient Cuff directly analyzes the harmfulness of the response to the potentially malicious query, isn't the maliciousness detection methods like LlamaGuard the more suitable baselines?
+ What is the trade-off with utility for the baseline methods? How to verify that Gradient Cuff reaches a better trade-off?

**Limitations:**

The authors discussed the limitations in Section 4.4 and 4.5 regarding the utility degradation and performance degradation when encountered with adaptive attacks. The broader impact is discussed in Section 6 saying that no negative impact of Gradient Cuff has been foreseen.

---

> ### Author Rebuttal · Authors · 2024-08-05
>
> **Question 1**: Is LlamaGuard the more suitable baselines?
>
> **R**: We thank the reviewer for pointing out this baseline for us. We use the Llama-Guard-2-8B as the classifier used in the LlamaGuard method and compare it with Gradient Cuff. The results are shown below.
> |Model|Method|FPR|TPR|
> |----------------|---------------|-------|-------|
> |vicuna-7b-v1.5|w/o defense|0.000|0.233|
> ||Gradient Cuff|0.034| 0.743|
> || LlamaGuard|0.040| 0.773|
> |llama2-7b-chat|w/o defense|0.000|0.614|
> ||Gradient Cuff|0.022|0.883|
> || LlamaGuard|0.040|0.760|
> From the table, we can conclude that:
>
> **(1)** They both keep a low FPR so both of the two won’t bring great utility degradation.
>
> **(2)** Gradient Cuff is superior to LlamaGuard in terms of defense performance against jailbreak attacks. Specifically, Gradient Cuff can achieve comparable TPR results on vicuna(0.743vs0.773) and much better results on llama2(0.883vs0.760).
>
> We need to emphasize that Llama-Guard-2-8B is an LLM based on LLaMA-2 and trained on massive data to learn to distinguish malicious queries, while our method and the other baselines do not use external models as guardrails (that is, only using the protected model to enhance safety). Though comparing to Llama-Guard-2-8B is unfair to Gradient Cuff, this result further corroborates the effectiveness of Gradient Cuff.
>
> Moreover, LlamaGuard needs to deploy two LLMs: the protected LLM itself and a LlamaGuard family model (at least 7B size), while Gradient Cuff only needs to deploy the protected LLM.
>
> We will add this discussion in the final version of this paper. It will provide more supportive evidence for Gradient Cuff’s effectiveness.
>
> ***
>
> **Question 2**: How to verify that Gradient Cuff reaches a better trade-off with utility?
>
> **R**: In section 4.1, we discussed the trade-off between TPR and FPR for Gradient Cuff and all the baselines, Figure 2 showed that our method can outperform PPL and SmoothLLM with a similar FPR and a much higher TPR.
>
> In section 4.5, we discussed the utility of the Gradient Cuff evaluated on the MMLU benchmark. The results showed that Gradient Cuff’s utility degradation is only brought by false refusal to some benign queries and the false refusal rate can be controlled by adjusting the FPR on the validation set.
>
> To answer the reviewer’s question, we computed all the baselines’ utility degradation following the same procedure introduced in section 4.5 and compared them with Gradient Cuff. The results are shown below
> | Model | Method | TPR | MMLU Accuracy |
> |---|---|---|---|
> | LLaMA | Gradient Cuff | 0.883 | 0.3780 |
> |  | SmoothLLM | 0.763 | 0.2706 |
> |  | PPL | 0.732 | 0.4107 |
> |  | wo/defense | 0.613 | 0.4517 |
> | Vicuna | Gradient Cuff | 0.743 | 0.4621 |
> |  | SmoothLLM | 0.413 | 0.3032 |
> |  | PPL | 0.401 | 0.4867 |
> |  | w/o defense | 0.233 | 0.4867 |
>
> From the results, we find that though SmoothLLM can achieve a very low FPR shown in Figure 2, it causes a dramatically large utility degradation because it has modified the original user query so it will damage the semantics of the user query. PPL attains the best utility and Gradient Cuff achieves the best performance-utility trade-off by
>
> **(1)** keeping the comparable utility with PPL
>
> **(2)** attaining a much higher TPR than the best baselines(e.g., 0.743vs0.413 on viucna) against jailbreak attacks.
>
> We thank the reviewer for this question and will add the utility comparison with baselines in section 4.5 of the revised version of this paper.

---

### Author Rebuttal · Authors · 2024-08-05

**We put some questions focusing on clarification in the General Reply**
***

**Question**: Could the author compare or discuss the proposed method against [2,3,4]?

**Response**: We thank the reviewer’s recommendation. We carefully read these 3 papers and found that these methods are all quite different from Gradient Cuff.

**(1)** Safety-aware decoding. For SafeDecoding[2], it finetuned an expert LLM using safety datasets and combined the token probability of both the expert LLM and the protected LLM during the decoding process.

**(2)** Trained classifier. For SPD[3], it trained an SVM classifier to detect jailbreaking inputs via the logit values.

**(3)** For RAIN[4], it proposed a novel inference method that allows the protected LLMs to evaluate their own generation and use the evaluation results to guide rewind and generation for AI safety.

RAIN needs multiple searches, rewinds, and exploration during the generation process, thus making it very time-consuming (much longer than Gradient Cuff according to our experience). The results in the paper also indicate that RAIN is not a powerful jailbreak defense method as it can only reduce the GCG's ASR to 72% though the GCG suffix they tested was only trained for 100 steps.  ([1] indicates that longer training steps make GCG[1] stronger and 100 steps is not very enough). We tested RAIN against our GCG jailbreak prompt (1000 steps training) on Vicuna and found that RAIN totally failed in defending jailbreaks. The results are shown below.

|  | GCG ASR |
|---|---|
| w/o defense | 1.000 |
| Gradient Cuff | 0.108 |
| RAIN | 1.000 |

We don’t test RAIN against other models and datasets as this method is very time-consuming and is not a powerful defense against jailbreak.

SPD doesn’t open-source the weights of the SVM model they trained, and the training cost is not affordable. They used thousands of jailbreak prompts (GCG PAIR AutoDAN) in the training dataset. The generation of the dataset needs huge time cost and money cost. So we choose to compare Gradient Cuff with another trained classifier instead - LLaMAGuard.

In conclusion, we compared Gradient Cuff with two new baselines: SafeDecoding and LLaMAGuard.  The results are shown below:

|  |  | FPR | TPR |
|---|---|---|---|
| vicuna | w/o defense | 0.000 | 0.233 |
|  | Gradient Cuff | 0.034 | 0.743 |
|  | llama guard | 0.040 | 0.773 |
|  | SafeDecoding | 0.280 | 0.880 |
| llama | w/o defense | 0.000 | 0.614 |
|  | Gradient Cuff | 0.022 | 0.883 |
|  | llama guard | 0.040 | 0.760 |
|  | SafeDecoding | 0.080 | 0.955 |

We can summarize the experimental results as follows:

**(1)** SafeDecoding can achieve the largest average TPR against jailbreak attacks. But the FPR of SafeDecoding is much higher which will bring unacceptable utility degradation.

**(2)** LLaMAGuard can achieve comparable TPR results with Gradient Cuff on vicuna and a much lower TPR performance on LLaMA.

The experimental results showed that Gradient Cuff can also outperform these two new baselines. Moreover, we should argue that these two new baselines all need to train a new model while Gradient Cuff only needs the protected LLM itself.

We think the discussion of comparing with these new baselines is interesting and valuable. We thank the reviewer again and decided to include this part in the revised version of the paper.

***

**Question**: Is it possible to apply the proposed defending method in closed-source GPT?

**Response**:  Thank the reviewer for this question. We adopt white-box settings because Gradient Cuff is actually for the developer of the model, who has full access to the model’s parameters including the embedding layer. We are currently also trying to make a new version of Gradent Cuff which uses character perturbation instead of embedding perturbation, thus the user who does not have full access to the model weights will also be able to implement the Gradient Cuff for closed-source LLMs like GPT-4.
***

**Question**: How to sample in a batch with the same temperature?

**Response**: This is a technical detail. The decoding is based on random sampling, so the items in the same batch will have different random processes during sampling. That means even if the items in a batch are identical, the response may be different.

We provide the sample code for batch inference here. In the sample code, even if all sentences in the prompt_list are identical, the responses to them may be different.

```python
def batch_inference(prompt_list, model_name):
   #prompt list contains multiple user query sentences
   model = AutoModelForCausalLM.from_pretrained(model_name).cuda()
   tokenizer = AutoTokenizer.from_pretrained(model_name)
   inputs=tokenizer(prompt_list,return_tensors="pt",padding=True)
   inputs={k:v.to(model.device) for k,v in inputs.items()}
   with torch.no_grad():
       input_ids = inputs["input_ids"]
       attention_mask = inputs["attention_mask"]
       prompt_length=input_ids.shape[-1]
       outputs = model.generate(
           input_ids = input_ids,
           attention_mask= attention_mask,
           do_sample = True
       )
       output_text = tokenizer.batch_decode(outputs[:,prompt_length:], skip_special_tokens=True)
   return output_text
```

***
**References**

[1] Universal and Transferable Adversarial Attacks on Aligned Language Models

[2] Safedecoding: Defending against jailbreak attacks via safety-aware decoding

[3] Single-pass detection of jailbreaking input in large language models

[4] RAIN: Your language models can align themselves without finetuning

---

### Decision · Program_Chairs · 2024-09-25

**Decision:**

Accept (poster)

**Comment:**

This paper introduces a new method, Gradient Cuff, for detecting jailbreaking attempts in language models by analyzing the refusal loss landscape. Overall, the reviewers find the developed method is novel, and the empirical results are promising. But meanwhile, several concerns are raised, regarding 1) the associated computational cost is high; 2) the evaluation of utility trade-offs is not sufficiently explored; 3) comparisons to additional baseline methods are needed; 4) some technique aspects need to be further clarified and the overall presentation should be further improved.

The rebuttal is considered, which addresses most of these issues (two reviewers raised their scores to 5: Borderline Accept), except the concern about costly computations. While this concern is critical, the AC believes addressing this efficiency issue is beyond the main scope of this paper (which aims to develop an effective strategy for detecting jailbreak attempts) and is legitimate to be left as future work. Given the overall positive positions from all reviewers, AC recommends accepting this submission.